# Research on Representative Volume Element Fex-Cy High-Temperature Mechanical Model Based on Response Surface Analysis

**Shining Lyu, Youshan Gao \*, Aihong Wang and Yiming Hu**

College of Mechanical Engineering, Taiyuan University of Science and Technology, Taiyuan 030024, China; b202112310025@stu.tyust.edu.cn (S.L.); 1996032@tyust.edu.cn (A.W.); huyiming1112@163.com (Y.H.)

\* Correspondence: taijixiaowangzi@foxmail.com or 2003011@tyust.edu.cn; Tel.: +86-1873-4834-200 or +86-1393-5173-225

**Abstract:** In this research, a multi-scale representative volume element method is introduced that combines the temperature and stress fields to analyze the force field distribution around microcracks in low-carbon steel using a combination of molecular dynamics and finite element analysis. Initially, an orthogonal experimental design was used to design the molecular dynamics simulation experiments. Next, a nano-level uniaxial tensile test model for mild steel was established based on the experimental design, and the uniaxial tensile behavior of low-carbon steel was investigated using molecular dynamics. Lastly, mathematical models of the modulus of elasticity E and yield strength Q of mild steel at a high temperature were obtained statistically using the response surface methodology. Meanwhile, a finite element model with a coupled temperature–stress field was established to investigate the force field distribution around the microscopic defects, and the microscopic crack stress concentration coefficient K was revised. The results indicate that regardless of the location of microcracks within the structure, the stress distribution due to size effects should be considered under high-temperature loading.

**Keywords:** molecular dynamics; multi-field coupling; representative volume elements; finite element

## 1. Introduction

During the production and processing of materials, the presence of machining defects is inevitable. These initial defects often fall into the category of microscopic imperfections that are difficult to observe with the naked eye. The mechanical distribution characteristics near these defects are crucial areas of study, especially in the context of large-scale structures such as mechanical and civil engineering projects. In recent years, there has been a growing body of research dedicated to the study of microdefects in alloys [1–4]. The research methods for studying microdefects have evolved from classical elastoplastic mechanics theories [5] to numerical approaches such as finite element analysis [6] and extended finite element methods [7]. This evolution has also seen a shift from idealized models to more complex ones [8,9]. In recent years, with the development of computational materials science, cellular automata [10] and molecular dynamics [11] have gradually been introduced into the study of microdefects. Due to the inherent differences between the microscale and macroscale properties of materials [12–14], there is currently a lack of comprehensive research that unifies the microscale properties of materials with their macroscopic performance.

Molecular dynamics methods, as a numerical method for detailed microscopic modeling at the molecular scale, have recently been widely applied to the research of mechanical properties of different types of crystalline materials [15–17]. Tang and Horstemeyer et al. studied the fatigue mechanical behavior of single magnesium crystals with different orientations using the MD method [18], and then, reviewed atomic simulation studies of microscopic crack extension in nickel and cooper [19].

It was shown that vacancies [20] in the microstructure as well as elemental composition [21] affect the calculation results, and there were potential influence rules among the factors, so the response surface method was used for the analysis. However, due to the huge number of molecular dynamics calculations and high hardware requirements, the orthogonal test method was used for the experimental design in order to reduce the number of simulation tests.

Metal structures often operate in complex environments and are subjected to varying temperatures, making it crucial to study their material properties under different temperature conditions. The high-temperature mechanical performance of steel requires extensive consideration [22–25]. However, previous research has mainly relied on practical macroscopic experiments, and there have been limitations in terms of economic feasibility and the macro–micro coupling perspective. Therefore, there is an urgent need to explore the material properties of carbon steel using molecular dynamics methods. To facilitate broader engineering applications, this paper aims to establish a mathematical model related to temperature, vacancy rate, and carbon content based on molecular dynamics simulation results.

Representative volume element, as a commonly used method for the research of the mechanics of inhomogeneous media [26], has been used to study the mechanical properties of multidirectional amorphous polymers using the RVE method by borrowing the continuum mechanics assumption [27], to construct representative volume elements considering inertial forces and body forces at the microscopic scale [28], and to conduct fatigue crack growth research on carbon steel by assessing representative volume elements. Temperature is one of the main factors affecting the mechanical properties of mild steel [29,30], yet there are few studies on microscopic cracking with coupled temperature–stress fields. Therefore, we proposed a multi-scale method based on representative volume elements to research microscopic crack force field distribution under the influence of temperature field, where finite element (FEM) and molecular dynamics (MD) approaches were used to research the macroscopic and microscopic crack force field distribution, respectively. Furthermore, the stress concentration factor for cracks was adjusted.

The research is organized as follows. Section 2 focuses on the basic theory of this research. Section 3 begins with a description of the technical route; then, it describes the design of the the simulation test using the orthogonal test method; then, it obtains the mathematical model of *E* and *Q* using molecular dynamics simulation and the response surface regression method; and finally, it establishes a crack finite element model to obtain the force field distribution. Section 4 further discusses the microscopic crack stress concentration factor. Section 5 provides the conclusion of this research.

## 2. Theory

Molecular dynamics calculations describe the position and momentum of each atom, and the equations of the motion of the atoms satisfy the Newtonian assumptions of motion, as in Equation (1). After time discretization, the complex continuous motion curves of individual atoms are transformed into discrete multiple linear motions.

$$m_i \boldsymbol{a}_i = \boldsymbol{f}_i = -\nabla U(\boldsymbol{r}_i) \tag{1}$$

in which $i$ denotes the sequence of atoms, and $m_i$, $\boldsymbol{a}_i$, $\boldsymbol{f}_i$, and $\boldsymbol{r}_i$ denote the mass, acceleration, interatomic force, and position of atom $i$, respectively. In this paper, the alloys are studied and the EAM action potential provides a favorable fit for the alloys. In the embedded atomic method calculation, the interatomic forces can be obtained using the following equation.

$$U(\boldsymbol{r}_i) = \sum_i U_{ij}(\boldsymbol{r}_i, \boldsymbol{r}_j) \tag{2}$$

$$\boldsymbol{f}_i = -\nabla U(\boldsymbol{r}_i) = 0, \ \boldsymbol{r}_{ij} > \boldsymbol{r}_{cut} \tag{3}$$

where $i$ and $j$ denote the sequence of atoms, $U_{ij}$ is the potential energy between atoms $i$ and $j$, $r_{ij}$ denotes the distance between atoms $i$ and $j$, and $r_{cut}$ denotes the truncation radius. The position and velocity calculation of atoms in large atomic systems are determined using the velocity Verlet calculation method [31], and the stress on the atom is calculated using Equation (4) [32].

$$\sigma_{xy} = \frac{1}{V^i} \sum_i \left[ \frac{1}{2} \sum_{j=1}^{N} (r_x^j - r_x^i) f_y^{ij} - m^i v_x^i v_y^i \right] \tag{4}$$

where the subscripts $x$ and $y$ denote Cartesian components, $V^i$ denotes the volume of atom $i$, and the superscripts $i$ and $j$ denote the atomic label. Atomic stress consists of both potential energy and kinetic energy.

For the purpose of describing the state of high stress around the defect, the stress concentration factor $K$ is adopted as an indicator to determine the size of the stress concentration. It is defined as the ratio of the maximum stress $\sigma_{max}$ around the defect to the nominal stress $\sigma$ at this point, as shown in Equation (5).

$$K = \frac{\sigma_{max}}{\sigma} \tag{5}$$

## 3. Methodology

### 3.1. Technology Route

A coupled RVE-based multi-scale temperature–stress field approach was used to simulate microscopic crack stress distributions at the microscopic and macroscopic scales. As shown in Figure 1, it was divided into two main parts: the FEM section and molecular dynamics computational models. The finite element part was used to provide boundary conditions (temperature and node displacement) to the microscopic calculations and slice the microscopic cracks nearby for the sub-model, and the microscopic model was used to obtain the microscopic material mechanical parameters $E$ and $Q$ via molecular dynamics calculations and update the sub-model material parameters. In order to extract the microscopic material mechanical parameters $E$ and $Q$ more quickly during the research, the orthogonal test method was used to design the simulation test, and the response surface regression model was used to obtain the mathematical models of $E$ and $Q$. The specific implementation process is shown in Figure 1.

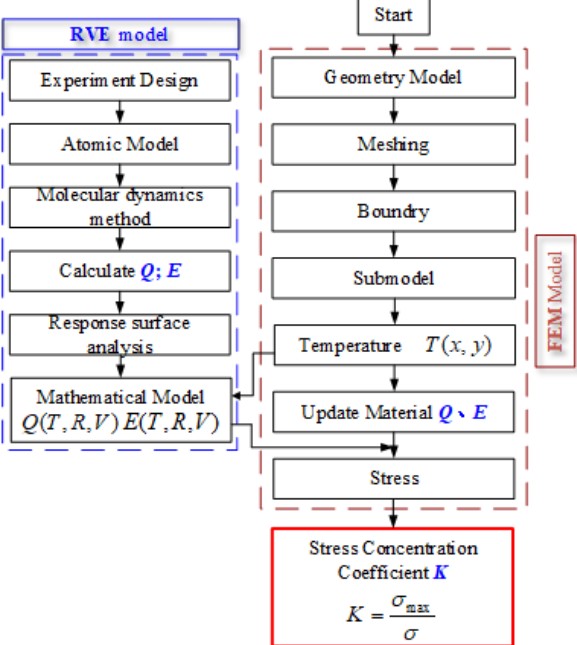

**Figure 1.** Methodology.

### 3.2. Experimental Design and Molecular Dynamics Simulation Model

The basic idea of response surface methodology is based on the idea of mathematical regression, which replaces complex models with regression models by fitting different points in the design space. This research used the OPTIMAL design method in Design-Expert for a more flexible experimental design. The carbon content of mild steel was about 1% to 2.2%, and the vacancy rate was about 1% to 5%, and the temperature range of the mechanical properties of mild steel observed in this paper was set between 300 K and 900 K. According to [33], the mechanical properties of mild steel deteriorate rapidly after the temperature exceeds 900 K, and it was not suitable to be used as the main support material anymore. The level grading of all factors was set using homogenization, the temperature T level was set to 4 levels, the C content R and void ratio V level were set to 5, the additional model points were set to 27, the lack of fit points were set to 10, and there were no replication points, for a total of 57 sets of experiments. The factor level hierarchical design and the corresponding values are shown in Table 1, and the results of the response surface test design are given in Table 2.

**Table 1.** Design of factors and levels.

| | Description | Level | | | | |
|---|---|---|---|---|---|---|
| | | **−2** | **−1** | **0** | **1** | **2** |
| $X_1$ | Temperature ($T$/K) | 300 | 500 | 700 | 900 | - |
| $X_2$ | C content ($R$/%) | 1 | 1.3 | 1.6 | 1.9 | 2.2 |
| $X_3$ | Vacancy ratio ($V$/%) | 1 | 2 | 3 | 4 | 5 |

**Table 2.** Experimental design of response surface method.

| Name | $X_1$ | $X_2$ | $X_3$ | Name | $X_1$ | $X_2$ | $X_3$ |
|---|---|---|---|---|---|---|---|
| 1 | 500 | 1 | 4 | 30 | 900 | 1 | 4 |
| 2 | 900 | 2.2 | 3 | 31 | 900 | 1.6 | 3 |
| 3 | 900 | 1.6 | 2 | 32 | 700 | 1.9 | 5 |
| 4 | 300 | 1 | 3 | 33 | 300 | 1.6 | 4 |
| 5 | 300 | 2.2 | 2 | 34 | 300 | 1 | 5 |
| 6 | 300 | 1 | 1 | 35 | 700 | 1.6 | 4 |
| 7 | 900 | 1.3 | 2 | 36 | 500 | 1.6 | 1 |
| 8 | 300 | 1.9 | 1 | 37 | 500 | 1.3 | 2 |
| 9 | 700 | 1.3 | 2 | 38 | 900 | 1 | 5 |
| 10 | 700 | 1.3 | 5 | 39 | 500 | 1.9 | 1 |
| 11 | 500 | 1.6 | 5 | 40 | 700 | 1.9 | 4 |
| 12 | 700 | 1.6 | 3 | 41 | 300 | 1.3 | 5 |
| 13 | 300 | 1.9 | 3 | 42 | 900 | 1 | 2 |
| 14 | 900 | 1.9 | 5 | 43 | 300 | 1.6 | 2 |
| 15 | 500 | 1 | 3 | 44 | 900 | 1.3 | 1 |
| 16 | 700 | 1.3 | 3 | 45 | 500 | 2.2 | 5 |
| 17 | 300 | 1.9 | 2 | 46 | 300 | 2.2 | 4 |
| 18 | 500 | 1.9 | 4 | 47 | 500 | 1.3 | 3 |
| 19 | 900 | 1.9 | 1 | 48 | 700 | 1 | 4 |

**Table 2.** *Cont.*

| Name | $X_1$ | $X_2$ | $X_3$ | Name | $X_1$ | $X_2$ | $X_3$ |
|------|-------|-------|-------|------|-------|-------|-------|
| 20 | 500 | 2.2 | 1 | 49 | 700 | 1.9 | 3 |
| 21 | 700 | 2.2 | 1 | 50 | 500 | 1.6 | 2 |
| 22 | 700 | 1 | 2 | 51 | 900 | 2.2 | 5 |
| 23 | 700 | 2.2 | 2 | 52 | 300 | 2.2 | 3 |
| 24 | 700 | 2.2 | 5 | 53 | 900 | 2.2 | 4 |
| 25 | 300 | 1.3 | 4 | 54 | 500 | 1.3 | 4 |
| 26 | 300 | 1.3 | 1 | 55 | 300 | 1.6 | 5 |
| 27 | 500 | 2.2 | 3 | 56 | 900 | 1.6 | 1 |
| 28 | 700 | 1 | 1 | 57 | 500 | 1.9 | 2 |
| 29 | 700 | 1.6 | 1 | - | - | - | - |

Simulations were performed according to the 57 sets of test cases in Table 2. Since the micromechanical properties are also related to the local atomic and vacancy distribution, 50 simulations were performed for each case to reduce the effect of this random relationship on the statistics of the material mechanical parameters, and the median of the respective Young's modulus as well, as the yield strength, were taken as the material parameters in that case.

Uniaxial stretching simulations of FeC alloys at the atomic scale were performed using a molecular dynamics approach. The FeC system was simulated using the open-source large-scale atomic/molecular parallel simulator LAMMPS, and the simulations were first performed with energy minimization, followed by 10 Ps relaxation in the NPT system, and finally, stretching in the micro-regular NVE system with temperature control using the velocity calibration method. The initial model had dimensions of 186.186 Å $\times$ 186.186 Å $\times$ 11.4576 Å with 34,000 atoms, Fe atom cell arrangement as BCC, and C atoms with vacancies randomly distributed in the Fe base. By varying the dimensions in the Y-direction of the box and stretching them on the *Y*-axis with a constant strain rate $\varepsilon_{yy} = 10^{10} \text{s}^{-1}$, the strain in the Y-direction can be expressed as the following equation.

$$\varepsilon_{yy} = \frac{l_y - l_{y0}}{l_{y0}} \tag{6}$$

The simulation time step was set to 0.001 ps, all three directions were set to the periodic boundary, and the number of simulation steps was 30,000. The present research was based on the research on Fe-C interatomic interactions by D.J. Hepburn [34], modified by Sebastien Garruchet, to obtain the Fe-C embedded atomic potential (EAM) to describe the interatomic interaction forces within Fe-C alloys. The batch simulation was performed using the Windows batching program "*.dat", visualized using Ovito, and the stress–strain data were output using the LAMMPS "fix print" function, and then, batch processed using matlab. As an example, the uniaxial stretching model of 300K1R1V (in Figure 2) was analyzed for the cell type during stretching, and it was found that the grains produced obvious dislocations during uniaxial stretching, varying significantly from 10 ps to 20 ps until 30 ps due to the large atomic spacing, which already struggled to maintain the bcc cell structure and was almost completely replaced by other cells (with atypical cell structures).

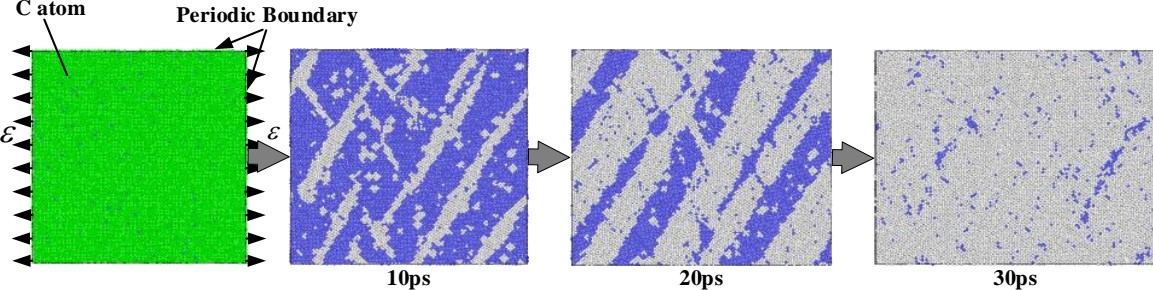

**Figure 2.** Simulation model.

During the micro-tensile testing of low-carbon steel in the range of 300–500 K, the material exhibited a clear linear elastic stage. At 300 K, after the linear elastic stage, the material would have an obvious yield stage, and then, entered the plastic stage. However, at high temperatures, the yield stage of the material in the range of 500–900 K was not significant, and with an increase in temperature, the yield stress of the material tended to decrease.

### 3.3. Mathematical Model and Variance Analysis

As shown in Figure 3, the mechanical properties of low-carbon steel were greatly influenced by the carbon content and vacancy ratio. In order to reduce the influence of local random relationships on the mechanical parameters of the material, 50 simulations were conducted for each case, and Young's modulus $E$ and yield strength $Q$ were calculated for each simulation. The median of the results of each calculation was taken as the mechanical characteristic of the material. The results are summarized in Table 3.

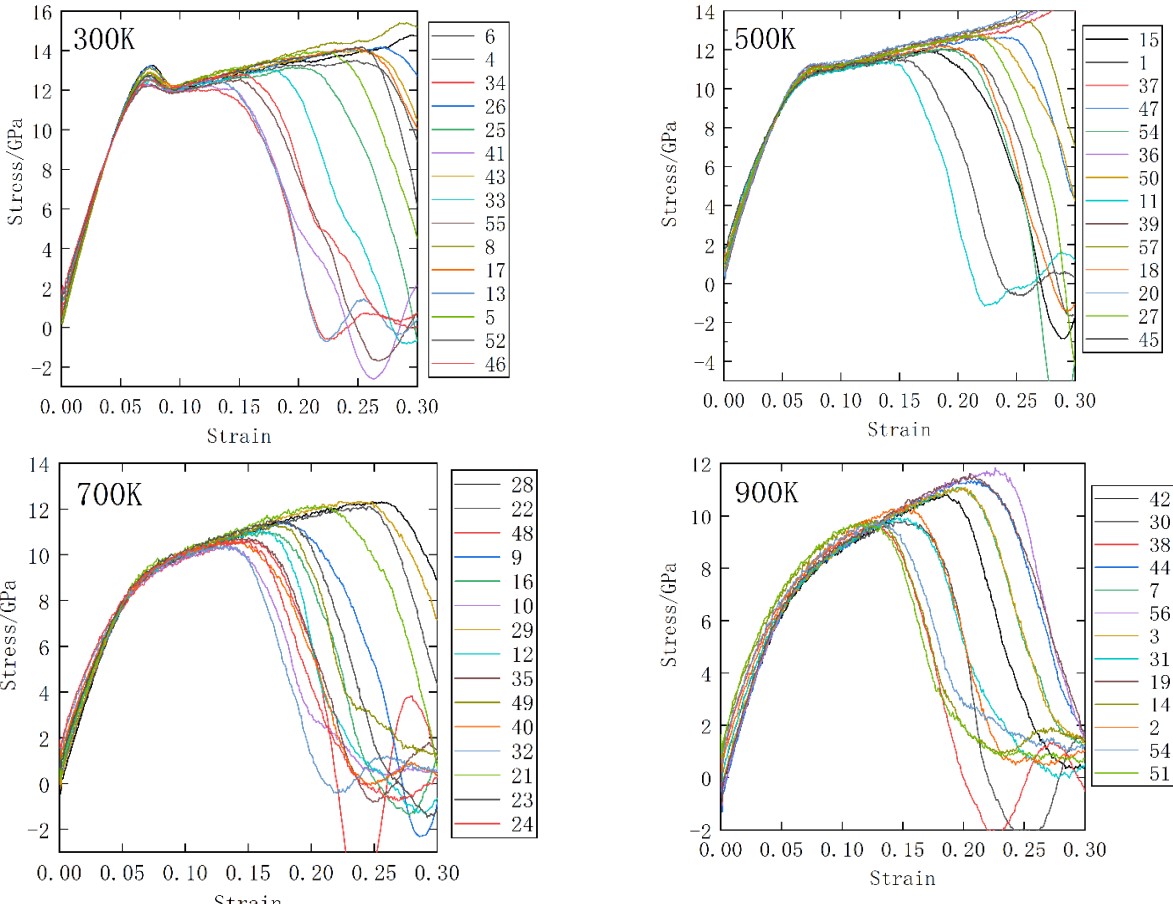

**Figure 3.** Curve of stress–strain.

**Table 3.** Simulation tests results.

| Name | E/GPa | Q/GPa | Name | E/GPa | Q/GPa | Name | E/GPa | Q/GPa |
|------|-------|-------|------|-------|-------|------|-------|-------|
| 1 | 157 | 10.8642 | 20 | 154 | 11.3220 | 39 | 152 | 11.2760 |
| 2 | 59 | 9.4343 | 21 | 86 | 10.0222 | 40 | 107 | 10.1762 |
| 3 | 46 | 9.5807 | 22 | 89 | 10.0143 | 41 | 211 | 12.2530 |
| 4 | 200 | 12.7287 | 23 | 92 | 10.2568 | 42 | 43 | 9.9332 |
| 5 | 214 | 12.8354 | 24 | 115 | 10.2299 | 43 | 205 | 12.8755 |
| 6 | 192 | 13.2412 | 25 | 208 | 12.4695 | 44 | 35 | 9.8690 |
| 7 | 43 | 9.7222 | 26 | 197 | 13.1815 | 45 | 173 | 10.9962 |
| 8 | 205 | 13.0851 | 27 | 164 | 11.1295 | 46 | 221 | 12.4089 |
| 9 | 90 | 10.0607 | 28 | 83 | 9.9673 | 47 | 155 | 11.0099 |
| 10 | 113 | 10.0334 | 29 | 85 | 10.0065 | 48 | 103 | 10.1257 |
| 11 | 169 | 10.8488 | 30 | 65 | 9.4247 | 49 | 100 | 10.3489 |
| 12 | 98 | 10.3705 | 31 | 56 | 9.4425 | 50 | 154 | 11.1332 |
| 13 | 213 | 12.6229 | 32 | 117 | 10.0845 | 51 | 88 | 9.7228 |
| 14 | 86 | 9.7070 | 33 | 213 | 12.4262 | 52 | 217 | 12.6140 |
| 15 | 153 | 10.9617 | 34 | 206 | 12.2664 | 53 | 72 | 9.4798 |
| 16 | 96 | 10.3152 | 35 | 105 | 10.1922 | 54 | 160 | 10.9185 |
| 17 | 210 | 12.8537 | 36 | 150 | 11.2414 | 55 | 215 | 12.2278 |
| 18 | 167 | 11.0399 | 37 | 151 | 11.1159 | 56 | 36 | 9.7911 |
| 19 | 37 | 9.2974 | 38 | 81 | 9.5083 | 57 | 157 | 11.1803 |

The experimental data in the table were subjected to regression analysis, and the response surface regression equations for $E$ and $Q$ were obtained as follows:

The following conclusions can be drawn from Tables 4–6. The F-values of the modulus of elasticity $E(T,R,V)$ and yield strength $Q(T,R,V)$ are 19,467.81 and 372.32, respectively, with $p$-values all less than 0.00001, so the model is significant and there is only a 0.01% probability that this model is inaccurate due to errors. Based on the $p$-value analysis, it can be concluded that the effect of temperature on the yield strength is more significant than the C content, as well as the vacancy rate, from the single factor analysis. As for Young's modulus, the temperature, C content, and vacancy rate all have a significant effect on it. The interaction factor of C content and vacancy rate does not have a significant effect on Young's modulus change, and the interaction of temperature with the other two factors often shows a more significant effect. The factors $TV$, $RV$, and $TRV$ will likewise have a significant effect on the yield limit.

$$
\begin{aligned}
E(T,R,V) &= 132.73991 + 0.481666T + 11.32157R + 8.9423V - 0.078182TR \\
&\quad -0.028533TV + 2.54469RV - 0.001214T^2 + 13.46899R^2 \\
&\quad -0.871564V^2 + 0.000522TRV + 0.000048T^2R \\
&\quad +0.000024T^2V + 0.000491TR^2 + 0.002053TV^2 \\
&\quad -0.340772R^2V - 0.217449RV^2 + 6.22653 * 10^{-7}T^3 \\
&\quad -2.90703R^3 + 0.042566V^3
\end{aligned}
\tag{7}
$$

$$
\begin{aligned}
Q(T,R,V) &= 19.4554 - 0.032261T + 1.15173R - 0.367567V + 0.005267TR \\
&\quad +0.001003TV - 0.163202RV + 0.000029T^2 - 1.4126R^2 \\
&\quad +0.010969V^2 + 0.000385TRV - 4.46411 * 10^{-6}T^2R \\
&\quad -1.04853 * 10^{-6}T^2V - 0.000382TR^2 + 9.40534 * 10^{-6}TV^2 \\
&\quad +0.003993R^2V - 0.003319VR^2 + 6.87185 * 10^{-9}T^3 \\
&\quad +0.319692R^3 - 0.002348V^3
\end{aligned}
\tag{8}
$$

**Table 4.** Young's modulus of compressive strength regression model.

| | Sum of Squares | Degrees of Freedom | Mean Variance | F-Value | *p*-Value | Significance |
|---|---|---|---|---|---|---|
| $E(T, R, V)$ | $1.94 \times 10^5$ | 19 | 10,197.12 | 19,467.81 | <0.0001 | Significant |
| $T$ | 16,112.46 | 1 | 16,112.46 | 30,761.08 | <0.0001 | Significant |
| $R$ | 35.92 | 1 | 35.92 | 68.58 | <0.0001 | Significant |
| $V$ | 265.85 | 1 | 265.85 | 507.54 | <0.0001 | Significant |
| $TR$ | 132.36 | 1 | 132.36 | 252.69 | <0.0001 | Significant |
| $TV$ | 914.05 | 1 | 914.05 | 1745.06 | <0.0001 | Significant |
| $RV$ | 3.66 | 1 | 3.66 | 6.99 | 0.0119 | |
| $T^2$ | 258.85 | 1 | 258.85 | 494.18 | <0.0001 | Significant |
| $R^2$ | 1.75 | 1 | 1.75 | 3.34 | 0.0755 | |
| $V^2$ | 21.88 | 1 | 21.88 | 41.77 | <0.0001 | Significant |
| $TRV$ | 0.2016 | 1 | 0.2016 | 0.3848 | 0.5388 | |
| $T^2R$ | 30.15 | 1 | 30.15 | 57.56 | <0.0001 | Significant |
| $T^2V$ | 87.22 | 1 | 87.22 | 166.51 | <0.0001 | Significant |
| $TR^2$ | 0.013 | 1 | 0.013 | 0.0248 | 0.8757 | |
| $TV^2$ | 27.9 | 1 | 27.9 | 53.27 | <0.0001 | Significant |
| $R^2V$ | 0.2625 | 1 | 0.2625 | 0.5012 | 0.4834 | |
| $RV^2$ | 1.11 | 1 | 1.11 | 2.12 | 0.1538 | |
| $T^3$ | 595.62 | 1 | 595.62 | 1137.13 | <0.0001 | Significant |
| $R^3$ | 0.9142 | 1 | 0.9142 | 1.75 | 0.1946 | |
| $V^3$ | 0.2686 | 1 | 0.2686 | 0.5128 | 0.4784 | |
| Residual | 19.38 | 37 | 0.5238 | | | |
| Total Variation Value | $1.938 \times 10^5$ | 56 | | | | |

**Table 5.** Yield limit of compressive strength regression model.

| | Sum of Squares | Degrees of Freedom | Mean Variance | F-Value | *p*-Value | Significance |
|---|---|---|---|---|---|---|
| $Q(T, R, V)$ | 79.88 | 19 | 4.2 | 372.32 | <0.0001 | Significant |
| $T$ | 3.32 | 1 | 3.32 | 293.64 | <0.0001 | Significant |
| $R$ | 0.0108 | 1 | 0.0108 | 0.9526 | 0.3354 | |
| $V$ | 0.0037 | 1 | 0.0037 | 0.327 | 0.5709 | |
| $TR$ | 0.0111 | 1 | 0.0111 | 0.9809 | 0.3284 | |
| $TV$ | 0.8068 | 1 | 0.8068 | 71.45 | <0.0001 | Significant |
| $RV$ | 0.0625 | 1 | 0.0625 | 5.53 | 0.0241 | Significant |
| $T^2$ | 3.51 | 1 | 3.51 | 310.68 | <0.0001 | Significant |
| $R^2$ | 0.0108 | 1 | 0.0108 | 0.9544 | 0.335 | |
| $V^2$ | 0.0135 | 1 | 0.0135 | 1.2 | 0.2807 | |
| $TRV$ | 0.1094 | 1 | 0.1094 | 9.68 | 0.0036 | Significant |
| $T^2R$ | 0.259 | 1 | 0.259 | 22.94 | <0.0001 | Significant |

**Table 5.** *Cont.*

|  | Sum of Squares | Degrees of Freedom | Mean Variance | F-Value | *p*-Value | Significance |
|---|---|---|---|---|---|---|
| $T^2V$ | 0.1599 | 1 | 0.1599 | 14.16 | 0.0006 | Significant |
| $TR^2$ | 0.0079 | 1 | 0.0079 | 0.6966 | 0.4093 |  |
| $TV^2$ | 0.0006 | 1 | 0.0006 | 0.0519 | 0.8211 |  |
| $R^2V$ | 0 | 1 | 0 | 0.0032 | 0.9552 |  |
| $RV^2$ | 0.0003 | 1 | 0.0003 | 0.0229 | 0.8805 |  |
| $T^3$ | 0.0725 | 1 | 0.0725 | 6.42 | 0.0156 | Significant |
| $R^3$ | 0.0111 | 1 | 0.0111 | 0.9791 | 0.3288 |  |
| $V^3$ | 0.0008 | 1 | 0.0008 | 0.0724 | 0.7894 |  |
| Residual | 0.4178 | 37 | 0.0113 |  |  |  |
| Total Variation Value | 80.3 | 56 |  |  |  |  |

**Table 6.** Reliability test and analysis of the model.

| Model | Standard Deviation | Average Value | Correlation Coefficient $R^2$ | Adjustment Coefficient ($R_a^2$) | Variation Coefficient (%) | Signal-To-Noise Ratio |
|---|---|---|---|---|---|---|
| $E(T, R, V)$ | 0.7237 | 132.83 | 0.9999 | 0.9998 | 0.5448 | 434.001 |
| $Q(T, R, V)$ | 0.1063 | 10.92 | 0.99448 | 0.9921 | 0.9734 | 59.977 |

Equations (7) and (8) suggest that, when $R$ and $T$ are constant, as $V$ increases, Young's modulus gradually increases due to an increase in the number of dislocations caused by vacancy inside the grains. However, the yield strength shows a decreasing trend as the defects increase. When $V$ and $T$ are constant, as $R$ increases, the stiffness of low-carbon steel gradually increases, and the stage of yield strength enhancement in comparison to Figure 3 becomes shorter, while the toughness decreases. When $R$ and $V$ are constant, as the temperature increases, the atomic activity intensifies, and the tendency of atoms to move away from each other becomes more apparent, resulting in a decrease in both the Young's modulus and yield strength of low-carbon steel. In order to verify the reliability of the equation and its statistical significance, a variance and confidence interval analysis can be performed.

Also, the correlation coefficients $R^2$ are 0.9999 and 0.99448, respectively, according to the model plausibility test analysis; The adjustment coefficients $R_a^2$ are 0.99998 and 0.9921, respectively; the signal-to-noise ratios are 434.001 and 59.977, respectively, both of which are much greater than 4, indicating that the regression model is highly reliable.

### 3.4. Temperature–Stress Field Coupling Finite Element Model

Taking the Q235 material as an example, a coupled temperature–stress field finite element model was established, as shown in Figure 4. A size $H \times L$ thin plate, where $H = 120$ mm and $L = 60$ mm, was placed on the bottom edge and at the center of the location of a circular hole representing a microdefect with a diameter of $2r = 50$ μm. On both sides of the thin plate, a line pressure of P = 300 N/mm was applied, respectively, and a temperature load of size T was applied on both sides of the thin plate, which lasted for 100 s. The ambient temperature was 22 °C and the convective medium was air. The surface of the thin plate was a convective surface, and the convective coefficient was 0.000056 w/mm². The submodeling technique was used to perform finite element simulation on the microscopic defects. A submodel with a size of $h \times l$ was cut from the thin plate, where $h = 0.25$ mm and $l = 0.5$ mm. The temperature obtained from the main

model calculation at the location was used as the ambient temperature for the submodel. The displacement and rotation calculated by the main model were mapped to the boundary elements of the submodel.

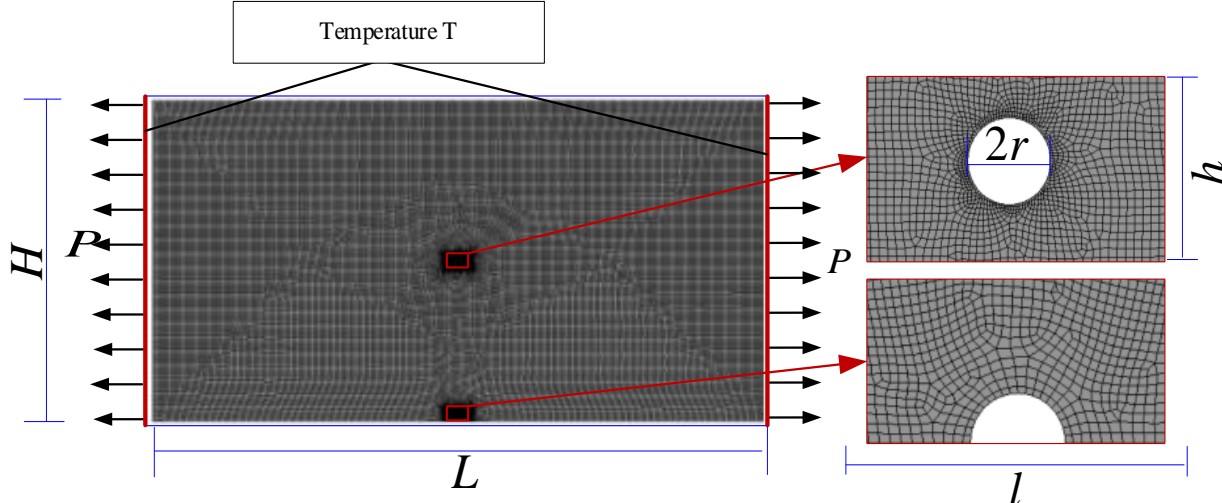

**Figure 4.** The model of FEM.

The parameters of the material in question are given in Table 7.

**Table 7.** Q235 material parameters.

| Temperature (K) | Density (g/cm³) | Elastic Modulus (GPa) | Poisson Ratio | Thermal Conductivity (w/mm²) |
|---|---|---|---|---|
| 300 | 7.8 | 203 | 0.3 | 0.000056 |
| 573 | 7.8 | 195 | — | 0.000048 |
| 673 | 7.7 | 183 | — | — |
| 773 | 7.7 | 169 | — | — |
| 873 | 7.6 | 126 | — | 0.000040 |
| 973 | 7.6 | 35 | — | — |

The boundary temperatures T of the thin plate were 300 K, 573 K, 623 K, 673 K, 723 K, 773 K, 823 K, and 873 K, respectively, and their corresponding stress cloud maps are shown in the Figure 5.

It can be seen from the stress distribution cloud map that the temperature distribution of the material is uniform at room temperature, the material properties are consistent, and the stress of the entire thin plate is 600 MPa. As the temperature load is applied, due to the uneven temperature of the thin plate and the inconsistent material properties, the stress at the upper and lower boundaries is lower, and the stress in the center is higher. Moreover, the stress gradually increases with the increase in temperature. As the temperature load increases to 823 K, the thin plate gradually becomes unstable, and the stress distribution gradually changes. Until the temperature rises to 873 K, the maximum stress value shifts from the center of the thin plate to both sides of the lower boundary of the thin plate.

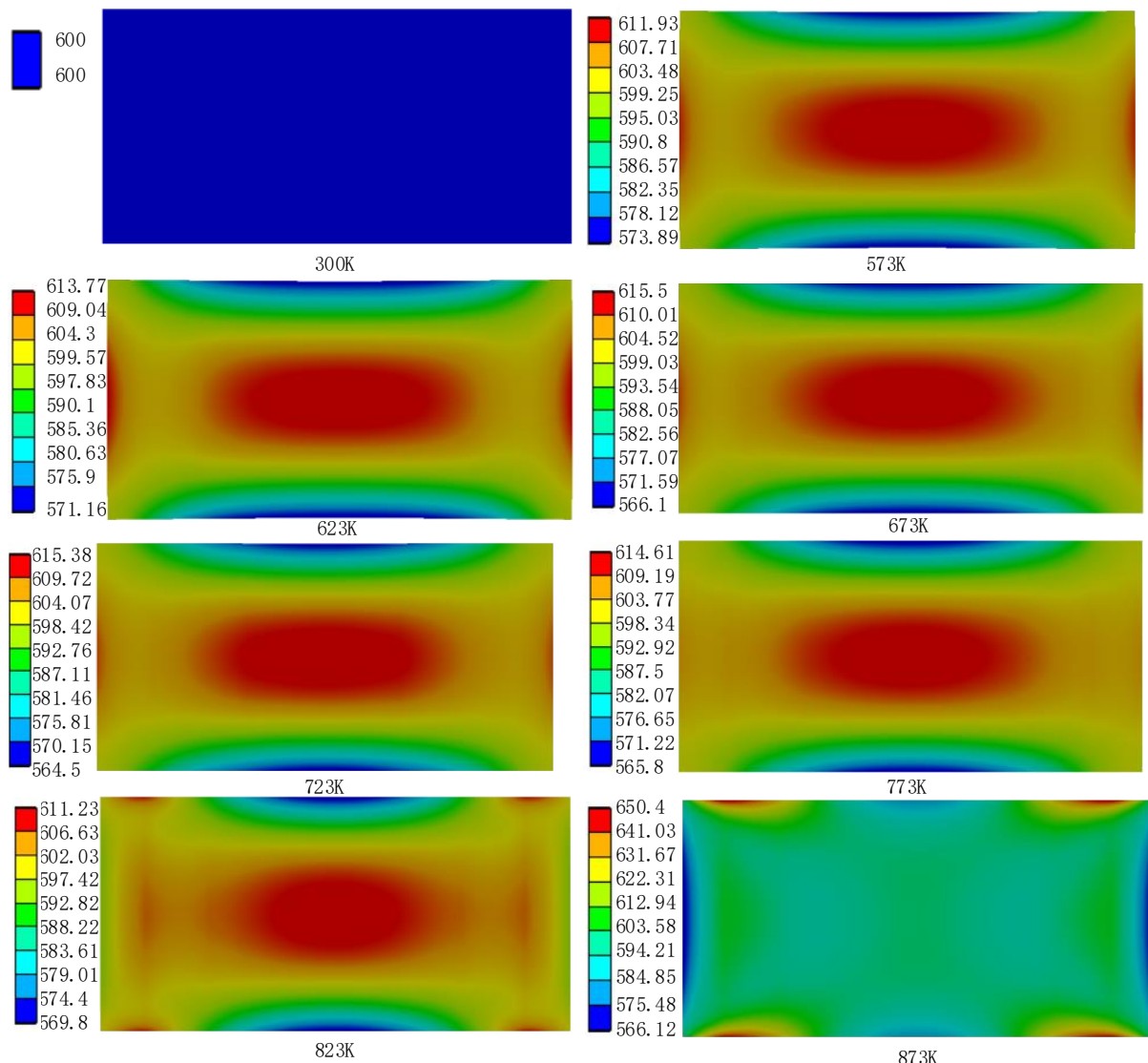

**Figure 5.** Plate stress cloud chart.

## 4. Microscopic Crack Coefficient Correction

Scholars believe that the stress concentration at the defect is actually smaller than the calculated stress due to the yield effect of the material causing stress redistribution near the defect. This explanation ignores the size effect of the mechanical properties of the material. According to the above analysis, the microscopic mechanical effects of the material are significantly different from the macroscopic effects. Yang's modulus at the micro and macro levels is almost the same at room temperature, but there is a significant difference at high temperatures. Due to fewer material defects at the micro-level, the yield limit is much higher than the measured value in a macroscopic situation. When the crack size is small, this size effect becomes more apparent, and therefore, it is necessary to correct the stress concentration coefficient of micro-cracks.

Based on what is described in Section 3.4, the stress concentration coefficient can be easily found without considering the size effect. In this section of the research, replacing the material elastic modulus in the sub-model and controlling the material elastic modulus with Equation (8), the C content of Q235 is about 1%, and a vacancy rate of 1% is used as an assumption for the solution. The controlling equation of the elastic modulus is shown in Figure 6, and Young's modulus gradually decreases with the increase in temperature. In the microscopic calculation, Young's modulus is higher than the macroscopic experimental

value at a high temperature; at room temperature, the calculated Young's modulus is slightly lower than the macroscopic experimental value.

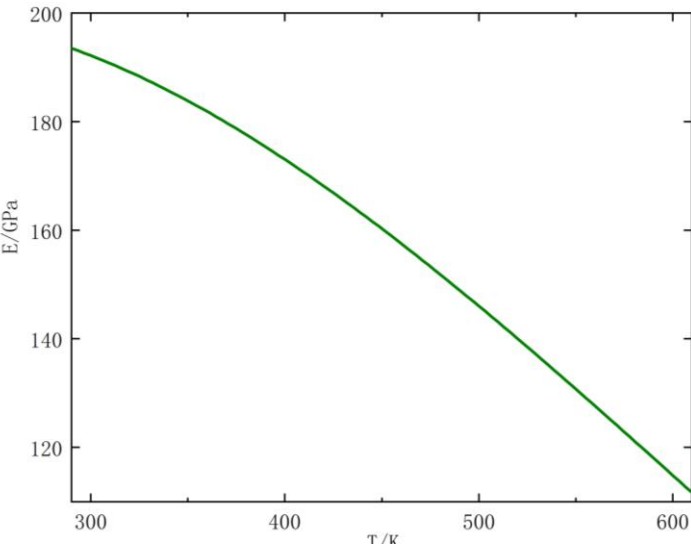

**Figure 6.** Thermal Young's modulus diagram.

Finite element calculations were performed for single-sided prototype notched tension and in-plane circular notched tension, respectively, and microscopic as well as macroscopic materials were used for comparison. The comparative stress clouds are shown in Figure 7. The first and third columns in the figure are the results of calculations without considering the microscopic size effect of the material, and the second and fourth columns are the results of calculations with considering the microscopic size effect of the material. The first and second columns are the cases of boundary cracks, and the third and fourth columns are the cases of central cracks.

At room temperature, the calculated stress distributions are relatively close because the microscopic and macroscopic material properties are close. When there is a high temperature load, this difference increases gradually with the increase in temperature load, and the microscopic calculation of the defect concentration stress is significantly smaller than the macroscopic material property calculation of the stress value, but the overall distribution trend is close. In other words, the size effect of the material is mainly reflected in the vicinity of the microscopic notch.

Since there are freer boundary conditions at the crack notch, there is a stress minimum in the tensile direction; the direction of the decreasing gradient of the stress maximum is perpendicular to the tensile direction, and at the same time, coincides with the crack expansion direction. The central crack has a greater concentrated stress because the central crack length is longer than the boundary crack. However, this does not mean that it is more likely to expand; expansion is also related to the shape factor.

The stress calculated in Section 3.4 was taken as the nominal stress, and the maximum value of microscopic defects calculated in this section is taken as the concentrated stress of defects and brought into Equation (5) to calculate the stress concentration factor $K_h$ and the modified stress concentration factor $K_w$. Considering that the concentrated stress $\sigma_k$ obtained from finite element analysis was less than 0.6 times the micro-yield strength $\sigma_s$ of the material, which had not reached the critical value for stress-plasticity redistribution, the crack size was not corrected. Taking circular defects as an example, the stress concentration factors of edge cracks and center cracks were calculated for a diameter of $2r = 50$ μm, and the concentration factors of microscopic cracks at different temperatures were obtained, as shown in Table 8.

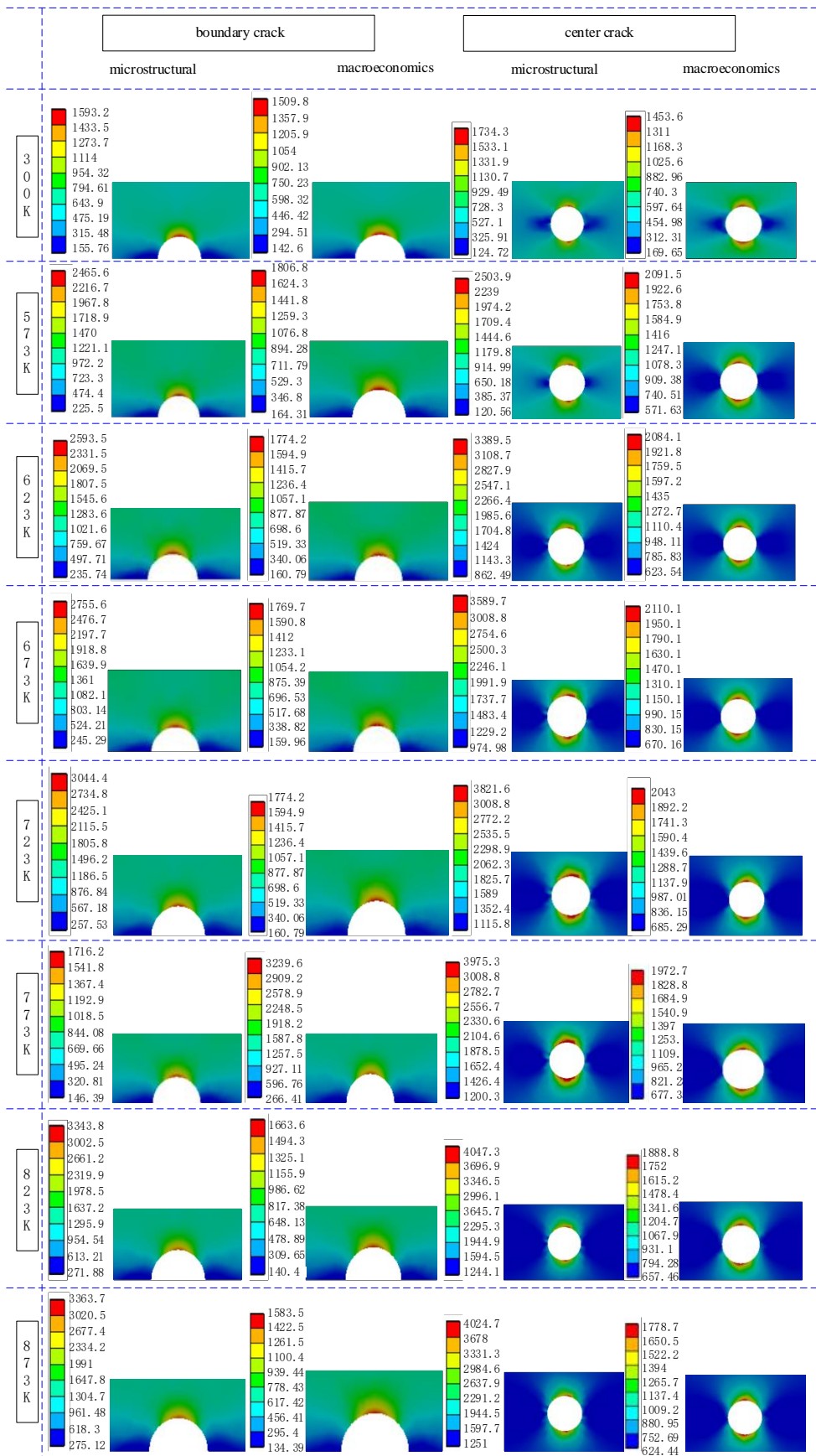

**Figure 7.** Cloud chart of crack stress.

**Table 8.** Boundary defect stress concentration coefficient.

| Temperature (K) | Maximum Stress (MPa) | Corrected Maximum Stress (MPa) | $K_h$ | $K_w$ |
|---|---|---|---|---|
| 300 | 1593.2 | 1509.8 | 2.655333333 | 2.516333333 |
| 573 | 2055.6 | 1661.9 | 3.578130168 | 2.892826681 |
| 623 | 2515.7 | 1774.2 | 4.100704179 | 2.89202582 |
| 673 | 2610.8 | 1716.1 | 4.600447569 | 3.023911473 |
| 723 | 2807.4 | 1662.1 | 4.946611693 | 2.928604151 |
| 773 | 2909.2 | 1547.4 | 5.13167875 | 2.729533788 |
| 823 | 3022.5 | 1555.4 | 5.304492804 | 2.72972973 |
| 873 | 2982.7 | 1453.1 | 5.110075554 | 2.489506416 |

Table 9 shows that the stress concentration factors considering the material micro-size effect are much smaller than those without considering the material size effect. The non-amended stress concentration factor gradually increases with temperature load from 300–773 K, the stress concentration factor of the central defect decreases, and then, increases at 823 K, and the stress concentration factor at the boundary increases, and then, decreases after continuing to increase at 823 K. The amended stress concentration factor shows a trend of increasing, and then, decreasing under temperature loads of 300–873 K. The stress concentration factor of the central defect reaches its maximum value at 623 K, and the stress concentration factor of the boundary defect reaches its maximum value at 673 K. Considering the material size effect, the micro-crack stress concentration factor does not change significantly with high temperatures.

**Table 9.** Center defect stress concentration coefficient.

| Temperature (K) | Maximum Stress (MPa) | Corrected Maximum Stress (MPa) | $K_h$ | $K_w$ |
|---|---|---|---|---|
| 300 | 1734.3 | 1453.6 | 2.8905 | 2.422666667 |
| 573 | 2503.9 | 2091.5 | 4.092342895 | 3.418321484 |
| 623 | 3389.5 | 2084.1 | 5.906389949 | 3.631658738 |
| 673 | 3589.7 | 2110.1 | 5.832832329 | 3.428659636 |
| 723 | 3821.6 | 2043 | 6.211761646 | 3.320763304 |
| 773 | 3975.3 | 1972.7 | 6.4680041 | 3.209677682 |
| 823 | 4047.2 | 1888.8 | 6.621402745 | 3.090162459 |
| 873 | 4024.7 | 1778.7 | 6.698566982 | 2.960404773 |

Based on the stress concentration factor and the yield strength of the micro-material, it can be inferred that the difficulty of micro-crack propagation is greater than that of macro-cracks, and the initiation stress of micro-cracks is greater than the initiation stress obtained in the statistical analysis of macro-materials. The micro-thermodynamic response of materials is significantly different from macroscopic phenomena. Therefore, the stress at the micro-crack site is redistributed, and the actual stress concentration factor of the micro-crack is not as large as that obtained via traditional calculation methods.

## 5. Conclusions

Through molecular dynamics and response surface methodology, we successfully established a mathematical model for the microstructural properties of low-carbon steel. The reliability of the model was validated through variance analysis. We proposed a multi-scale representative volume element method that considers temperature–stress coupling to

address stress redistribution caused by material size effects. In this study, we focused on two microcrack scenarios, namely, material boundary cracks and material center cracks, and investigated their stress distributions.

This research also considered the coupling effects of temperature and stress, as well as the material size effect, and made corrections to the stress concentration factor of microcracks. It is worth noting that, when considering the size effect, the corrected stress concentration factor under high-temperature loads is significantly smaller than the uncorrected results. With increasing temperature loads, the correction effect becomes more pronounced. Under normal temperature conditions, traditional calculation methods are effective for micro-stress calculations in engineering applications. However, at high temperatures, it is necessary to consider the size effect for correction.

The results of this study provide valuable insights into the high-temperature mechanical behavior of low-carbon steel materials. And useful references are also provided for future engineering applications. The proposed approach and models can be employed for material design optimization and crack analysis to ensure material performance stability under various temperatures and loads.

**Author Contributions:** Data curation, A.W.; Funding acquisition, S.L., Y.G. and A.W.; Methodology, S.L. and Y.G.; Software, S.L.; Writing—Original Draft, S.L.; Writing—Review and Editing, A.W. and Y.H. All authors have read and agreed to the published version of the manuscript.

**Funding:** This research was funded by the Shanxi Provincial Natural Science Foundation, China (grant number 20210302123217), the Innovation Project Fund for Graduate Students of Shanxi Province, China (grant number 2023KY639), the Research Project Supported by the Shanxi Scholarship Council of China (grant number 2020-124), the Innovation Project Fund for Graduate Students of Taiyuan University of Science and Technology, China (grant number BY2022014), and the Scientific Research Project of the China Three Gorges Corporation (grant number Z212202019).

**Institutional Review Board Statement:** Not applicable.

**Informed Consent Statement:** Not applicable.

**Data Availability Statement:** Data is contained within this article.

**Acknowledgments:** We would like to thank Shanxi Center of Technology Innovation for Electrohydraulic Control and Health Management of Heavy Machinery for funding this research. We thank Li Yajun for his contribution to the data processing part of this research.

**Conflicts of Interest:** The authors declare no conflict of interest.

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
