# Peer review of "Research on Representative Volume Element Fex-Cy High-Temperature Mechanical Model Based on Response Surface Analysis"

_applsci, doi:10.3390/app132011531_

Round 1

Reviewer 1 Report

In this study, the mechanical properties of low carbon steel material were examined in detail using the representative volume element method.

The literature research should be deepened in the introduction part. It should be stated how the study differs from other studies.

The chemical composition of the material used and its basic mechanical properties at room temperature should be given.

The results should be written in clear bullet points.

Reviewer 2 Report

Manuscript about a mathematical model for micro-material coefficients of low carbon steel by the molecular dynamics and response surface methodology, by multiscale representative volume element method is proposed, which is considered the stress redistribution caused by material size effect. The research on force fields near microscopic defects is extremely important. The authors have done a great job. But there are a few notes:

1. Fig.4 needs to be enlarged.

2. In Table 3, the elastic modulus values must be rounded to whole numbers.

3. In fig.5 the numbers on the scales are hard to see. The drawing needs to be enlarged.

4. Enlarge the labels of the axes in Fig. 6.

5. Fig.7 could be improved for better understanding and comparison

Reviewer 3 Report

There is a vague sentence “propose” in the abstract section. Also, this section only introduces the study and method used, but findings, suggestions, or conclusion statements are missing.

The novelty statement for this study is not clear. The proposed method is to achieve what? The aim and focus of this study must be stated clearly.

Under section 3.2. What is the vacancy rate??              

The results section was like a report. The mechanism that governs the reported results was missing. Also the results were not compared with another study for validation.

What is the meaning of “name” used on the tables?

What is the experimental error percentage of this study?

In conclusion: Is this study proposing a model or investigating it?

There is a vague sentence “propose” in the abstract section. Also, this section only introduces the study and method used, but findings, suggestions, or conclusion statements are missing.

The novelty statement for this study is not clear. The proposed method is to achieve what? The aim and focus of this study must be stated clearly.

Under section 3.2. What is the vacancy rate??              

The results section was like a report. The mechanism that governs the reported results was missing. Also the results were not compared with another study for validation.

What is the meaning of “name” used on the tables?

What is the experimental error percentage of this study?

In conclusion: Is this study proposing a model or investigating it?

Round 2

Reviewer 3 Report

The first sentence in the abstract with 'proposed' must be rephrased.

The reference is inconsistent. 

The first sentence in the abstract with 'proposed' must be rephrased.

The reference is inconsistent. 

Author Response

Dear Reviewer,

Thanks for your comment concerning our manuscript entitled “Research on Representative Volume Element Fex-Cy High Temperature Mechanical Model Based on Response Surface Analyzation”. We have studied your comments carefully and have made correction which we hope meet with approval.

Responds to the reviewer’s comments:

Response to comment1 :

The first sentence in the abstract with 'proposed' must be rephrased.

We have made revisions to the abstract section, which is as follows:

In this research, a multi-scale representative volume element method is introduced that combines temperature and stress fields to analyze the force field distribution around microcracks in low carbon steel using a combination of molecular dynamics and finite element analysis. Initially, an orthogonal experimental design was used to design the molecular dynamics simulation experiments. Next, a nano-level uniaxial tensile test model for mild steel was established based on the experimental design, and the uniaxial tensile behavior of low carbon steel was investigated by using molecular dynamics. Lastly, the mathematical models of the modulus of elasticity E and yield strength Q of mild steel at high temperature were obtained statistically using the response surface methodology. Meanwhile, a finite element model with coupled temperature field-stress field was established to investigate the force field distribution around the microscopic defects, and the microscopic crack stress concentration coefficient K was revised. The results indicate that regardless of the location of microcracks within the structure, the stress distribution due to size effects should be considered under high-temperature loading.

Response to comment2 :

The reference is inconsistent.

We have revised reference.